# Evaluation of the Antibacterial Effect of Aurone-Derived Triazoles on *Staphylococcus aureus*

**DOI:** 10.3390/antibiotics12091370

**Published:** 2023-08-26

**Authors:** Csilla Klara Szepe, Arjun Kafle, Shrijana Bhattarai, Scott T. Handy, Mary B. Farone

**Affiliations:** 1Department of Biology, Middle Tennessee State University, Murfreesboro, TN 37132, USA; cs5r@mtmail.mtsu.edu; 2Department of Chemistry, Middle Tennessee State University, Murfreesboro, TN 37132, USAscott.handy@mtsu.edu (S.T.H.)

**Keywords:** aurone, triazole, *Staphylococcus*, MRSA

## Abstract

Infections caused by antibiotic-resistant bacteria continue to pose a significant public health threat despite their overall decreasing numbers in the last two decades. One group of compounds fundamental to the search for new agents is low-cost natural products. In this study, we explored a group of newly synthesized novel aurone-derived triazole compounds to identify those with pharmaceutical potential as inhibitors of antibiotic-resistant *Staphylococcus aureus*. Using the broth microdilution method, antibacterial activities against methicillin-resistant *S. aureus* ATCC 43300 (MRSA) and methicillin-sensitive *S. aureus* ATCC 29213 (MSSA) were identified for four aurone-derived triazole compounds, AT106, AT116, AT125, and AT137, using the half-maximal inhibitory concentrations for the bacteria (IC_50_) and mammalian cell lines (CC_50_). Compounds AT125 and AT137 were identified to have pharmaceutical potential as the IC_50_ values against MRSA were 5.412 µM and 3.870 µM, whereas the CC_50_ values measured on HepG2 cells were 50.57 µM and 39.81 µM, respectively, resulting in selectivity indexes (SI) > 10. Compounds AT106 and AT116 were also selected for further study. IC_50_ values for these compounds were 5.439 µM and 3.178 µM, and the CC_50_ values were 60.33 µM and 50.87 µM, respectively; however, SI values > 10 were for MSSA only. Furthermore, none of the selected compounds showed significant hemolytic activity for human erythrocytes. We also tested the four compounds against *S. aureus* biofilms. Although AT116 and AT125 successfully disrupted MSSA biofilms, there was no measurable potency against MRSA biofilms. Checkerboard antibiotic assays to identify inhibitory mechanisms for these compounds indicated activity against bacterial cell membranes and cell walls, supporting the pharmaceutical potential for aurone-derived triazoles against antibiotic-resistant bacteria. Examining structure–activity relationships between the four compounds in this study and other aurone-derived triazoles in our library suggest that substitution with a halogen on either the salicyl ring or triazole aryl group along with triazoles having nitrile groups improves anti-Staphylococcal activity with the location of the functionality being very important.

## 1. Introduction

*Staphylococcus aureus* and its antibiotic-resistant strains are well-known in medical and research communities. They are not only capable of causing illnesses with significant mortality [1,2], but with an annual hospitalization of 300,000 in the US alone [3], these pathogens, including methicillin-resistant *S. aureus* (MRSA), continue to pose a notable public health threat [4,5,6,7]. In addition to its healthcare system presence (HA-MRSA), MRSA strains can also be community-associated (CA-MRSA) and livestock-associated (LA-MRSA) [4,8,9]. Although these MRSA strains have been traditionally characterized as genetically distinct, interaction of individuals from the community with livestock and with those in healthcare environments has resulted in the intermingling of these strains [4,8,10], and although LA-MRSA strains do not typically carry the genes for virulence and antibiotic resistance found in HA- and CA-MRSA, the ability to acquire these genes could lead to new strains with increased virulence in humans [10,11]. Furthermore, *S. aureus* is capable of forming multilayered cell masses called biofilms that increase its ability to resist antibiotics, generating a persistent source for recurring infections that are often associated with increased mortality and morbidity [12,13].

Biofilm development by *S. aureus* is generally a multi-stage process that begins with the attachment of planktonic (free-floating) cells to biotic or abiotic surfaces using cell wall-anchored proteins, cell wall teichoic acids, and extracellular DNA (eDNA) [12,13,14,15,16,17]. This adhesion may also be dependent upon the interaction of hydrophilic bacterial adhesins with hydrophobic surfaces [12,13,14,15]. In the second stage of biofilm development, the attached *S. aureus* cells begin dividing, forming microcolonies and a matrix of extracellular polymeric substance (EPS), consisting primarily of polysaccharides, proteins, or eDNA [16]. As the bacterial cells continue to multiply, the *S. aureus* biofilm enters a maturation and re-structuring stage [12,13,16]. This mature biofilm is characterized by a tower-like structure with channels for nutrient transport and waste removal that develop as a result of early dispersal of some of the *S. aureus* cells [12,13,15,16]. Bacteria within the tower exhibit greater antibiotic resistance as well as differences in cell size, growth, and metabolism [14]. Dispersion of biofilm is likely a multi-stage process that begins with dispersal for channel formation and is controlled by the Agr quorum sensing system [12,13,15,16]. Final dispersion and detachment of bacterial cells are mediated by surfactant-like phenol-soluble modulins (PSMs) that break non-covalent bonds within the biofilm as well as by nucleases and proteases [12,13,15,16]. These now planktonic cells may contribute to bacteremia in a patient or attach to another site and begin the formation of new biofilm; the remaining EPS matrix from a dispersed biofilm may serve as a site for recolonization of a surface thus contributing to the persistence or recurrence of infections [12,16,17].

In addition to the threat that *S. aureus* poses, there are several additional factors that warrant the search for new antimicrobial agents. One key factor is that resources for research into new agents have significantly shrunk since the 1990s as large pharmaceutical companies have completely abandoned the field [18], resulting in the current environment in which the introduction of new therapeutics lags behind the need for them [19]. On the other side of the balance sheet are the growing costs of treating *S. aureus*-related infections. In the US alone, HA-related infections generated an added burden of $10 billion or more annually [20,21], and recent reports from other parts of the world also show comparably high costs of treating patients for these infections [22,23,24,25]. This environment emphasizes the critical necessity for new, low-cost antibacterial agents.

One source for potential agents with bioactivity against *S. aureus* is natural products, especially aurones (2-benzylidene-1-benzofuran-3(2*H*)-ones), which is a flavonoid subfamily that contributes to the yellow coloring of some flowering plants [26]. These small molecules, both natural and synthetic, have been reported to inhibit infectious agents, including bacteria [26,27,28,29,30,31]. Due to the bioactivity of these compounds, efforts to synthesize aurone-derived compounds of high concentration and purity using efficient, inexpensive, and environmentally friendly methodologies are evolving [32,33]. In an attempt to further diversify the aurones by synthesizing azido-substituted aurones via copper-catalyzed azidation, a new class of aurone-derived salicyl-substituted 1,2,3-triazoles was produced [34]. An aurone-derived triazole library of 42 compounds was subsequently screened for bioactivity. Here we report the antibacterial activity of four aurone-derived triazoles against methicillin-sensitive *S. aureus* (MSSA) and MRSA along with toxicity and phenotypic studies that demonstrate the potential of these molecules for future pharmaceutical use.

## 2. Results

### 2.1. Aurone-Derived Triazole Anti-Staphylococcal Activity and Toxicity for Mammalian Cells

After screening the library of AT triazoles against *S. aureus*, four were chosen for further study (Figure 1). Table 1 and Appendix A depict the computed half-maximal concentrations as well as the maximal selectivity index (SI) ratios. All four AT compounds showed inhibition against MSSA with relatively low IC_50_ concentrations, but only AT125 and AT137 were able to additionally inhibit MRSA with concentrations below 6 µM. Although AT106 exhibited less toxicity for the HepG2 and HeLa cells, such that higher concentrations (above 60 µM) of AT106 were less toxic compared to the other three AT compounds, higher concentrations of AT106 were required for anti-Staphylococcal activity. Conversely to their antibacterial activity where AT137 was the strongest and AT106 was the weakest performer, AT106 is the least toxic with CC_50_ concentrations above 60 µM whereas AT137 is the most toxic with a 2.838 µM CC_50_. Still, all four compounds show pharmaceutical potential as they all have selectivity towards *S. aureus*. Unfortunately, these initial tests also show that only AT137 and AT125 were able to meet our selectivity criteria against MRSA, but we retained AT106 and AT116 for further testing as they could still be workable in mixtures with other bioactive compounds.

In addition to the toxicity for mammalian cells, we assessed the hemolytic activity of these AT compounds against human red blood cells. The results of the initial hemolysis assay in Table 2 show that all but AT125 have no hemolytic activity at their half inhibitory concentrations, and all of them show less than 5% at 50 µM, confirming that these ATs appear to be safe for delivery through the bloodstream of patients.

### 2.2. Bacterial Growth in Aurone-Derived Triazoles

Growth curve experiments were conducted to gain insight into the dynamics of the inhibitory effect of the AT compounds. The graphs in Figure 2A–H show that all ATs successfully inhibited bacterial growth during the two-day period in the two highest AT concentrations (25 µM and 100 µM). In concentrations closer to the measured IC_50_ values (5–25 µM), the graphs reflect our earlier distinction between strong inhibitors (AT125, AT137) and weaker inhibitors (AT106, AT116) of *S. aureus* growth. These results signify that AT125 (Figure 2E,F) and AT137 (Figure 2G,H) are successful in blocking bacterial growth at concentrations below 10 µM.

### 2.3. Biofilm Inhibition Microscopy

In addition to their inhibition potential against planktonic bacteria, the ATs were further investigated for disruption of *S. aureus* biofilms as these masses of bacteria pose a greater risk to patients [12,13,35]. Biofilm disruption was assessed by quantifying changes in bacterial cell numbers and the percentages of live and dead cells compared to the mock-treated control. As the imaging results in Figure 3 and Appendix A show, AT116 and AT125 treatments resulted in increased numbers of dead bacteria (2.42–3.95 fold increases) in pre-formed MSSA biofilms. All AT treatments resulted in 0.24–0.93 fold reductions of total bacterial cell numbers compared to the mock-treated control. AT116 and AT125 were the only two compounds with both reductions in total bacterial cell numbers and increases the dead cells. AT125 produced the largest fold increase in dead cells whereas AT137 had the largest decrease in total bacterial cells (Figure 3, Appendix A). Similar treatment and imaging were conducted for MRSA, but none of the ATs showed significant potency against MRSA biofilms with this assay.

### 2.4. Checkerboard Assay

As an initial step to determine potential modes of action for the ATs, checkerboard assay experiments were conducted wherein mixtures of ATs and antibiotics used against *S. aureus* were tested for synergistic or antagonistic relationships depending on the fractional inhibitory concentration index (FICi) shown in Table 3. All ATs, except AT125, showed synergy with daptomycin against MSSA and MRSA as their FICi were below 0.5. AT125 was only synergistic with daptomycin against MSSA. AT116 and AT125 also showed synergy with vancomycin against MSSA. However, none of the compounds showed interaction with vancomycin against MRSA. Further testing with additional antibiotics and phenotypic assays will be necessary to further assess potential modes of action.

## 3. Discussion

Our current results extend the existing set of evidence of the bioactivity of aurones by using aurone-derived triazoles with activity against *S. aureus*, including MRSA. The increasing resistance of microbes to existing therapies necessitates improved antimicrobials, and triazole compounds are well-established molecular scaffolds for pharmaceuticals, including antimicrobials [3,36,37,38]. The 1,2,4-triazoles have been used as antifungal agents since the discovery of this activity in the 1940s [37,39]. The 1,2,4-triazoles possess additional bioactivities and are used therapeutically as antibacterial, antiviral, anticancer, anticonvulsant, antidiabetic, and anti-inflammatory agents, with additional activities as hypnotics, antidepressants, migraine treatments, and skeletal muscle relaxants [40,41,42,43,44,45]. Due to this range of bioactivity and the ease with which triazoles bind diverse chemical species, their popularity as a pharmaceutical scaffold has increased, particularly for 1,2,3-triazoles which have become the most frequently studied motif among the azoles [37,46]. The 1,2,3-triazoles have also shown promise as anti-infectives with reported activity against bacteria, viruses, fungi, and parasites [46,47,48,49,50,51].

In this report, we have described the anti-Staphylococcal activity of aurone-derived 1,2,3-triazoles (ATs) formed by the reaction of aurones with sodium azide. The resulting ATs possess two sites for alkylation chemistry and the potential for modification of the ring system. To assess the pharmaceutical potential of the AT compounds, we used the standard method of calculating their selectivity index (SI), which is the ratio of their half-maximal toxicity concentration and their half-maximal inhibitory concentration. An SI higher than 10 shows that the compound has a large enough selectivity towards the bacterium, but not mammalian cells. The AT compounds chosen for this study (AT106, AT116, AT125, and AT137) all had SIs of >10 with mammalian cell lines and were not hemolytic for human erythrocytes indicating efficacy for delivery in the bloodstream. All four of the ATs were effective against MSSA, exhibiting IC_50_ values between 3.09–5.44 µM (1.08–1.63 µg/mL). These concentrations are comparable to MIC ranges and reported IC_50_ values for the reference antibiotics vancomycin and linezolid [52,53,54,55]. IC_50_ values of the ATs for MRSA were between 3.87–22.38 µM (1.35–6.71 µg/mL) with only AT125 and AT137 exhibiting IC_50_ values less than the MICs for vancomycin and linezolid. The inhibition of MRSA by AT125 and AT137 is confirmed by the 48 h growth curves where only concentrations lower than the IC_50_ values allowed for the growth of MRSA.

Biofilms associated with *S. aureus* infections can be difficult to treat and eliminate due to multiple factors, including impermeability associated with chemical features of a drug or the physiology of the bacterial cells in the biofilm, such as reduced proliferation or metabolism, especially if dormant bacteria known as persisters are present [12]. For this reason, we treated pre-formed biofilms of MSSA or MRSA with the four AT compounds. Disruption of biofilm was assessed using the LIVE/DEAD BacLight^TM^ assay which contains the dyes SYTO9 and propidium iodide. SYTO9 is transported across intact cell membranes and stains the nucleic acids, resulting in the green fluorescence of living cells. Propidium iodide cannot cross membranes of live cells, thus only staining nucleic acids of dead cells, producing a red fluorescence. Comparisons of live and dead cell numbers, as well as total bacterial cell numbers, to mock-treated control biofilms, were used to assess the efficacy of AT treatment. All ATs were effective against MSSA using the 100 μM concentration. This concurs with other observations that biofilms are more resistant to antibiotics, requiring concentrations up to 1000 times greater than required for planktonic bacteria [56].

Although the mode of action for these AT compounds is not yet clear, the synergism of ATs observed with vancomycin in checkerboard assays suggests the *S. aureus* cell wall as a potential target. *S. aureus* cell walls are composed of peptidoglycan, which is a layered polymer of alternating *N*-acetylmuramic acid (MurNAc) and *N*-acetylglucosamine (GlcNAc) molecules connected at the MurNAc residues by a short chain of amino acids [57]. The main EPS constituent of *S. aureus* polysaccharide biofilms is poly-*N*-acetylglucosamine (PNAG), also referred to as polysaccharide intercellular adhesin (PIA) [12]. If the ATs target cell wall polysaccharide synthesis, it is possible that they might also target the synthesis of biofilm polysaccharides. The antifungal triazole compound, itraconazole, has been shown to disrupt *S. aureus* biofilm, and although it blocks fungal ergosterol biosynthesis, it was proposed that the itraconazole might also target bacterial polysaccharide synthesis as does another antifungal drug, caspofungin [58]. Another study identified antifungal azoles as having *S. aureus* antibiofilm activity when combined with the antibiotic tetracycline. In this study, the authors demonstrate that the azoles act as competitive inhibitors of active efflux pumps in *S. aureus* biofilms, thus maintaining higher concentrations of the antibiotic in the bacterial cells by blocking the exit of the antibiotic [59]. This could also support the AT synergy observed with vancomycin or daptomycin in our study. An aurone-derived compound with anti-*S. aureus* biofilm activity suggests yet another potential mechanism of action for the ATs by the disruption of the quorum sensing [60]. The ATs also have the advantage of being small molecules (all <370 Da) which facilitates biofilm penetration [56]. Although all the AT compounds reduced total bacterial cell counts and decreased the numbers of viable bacterial cells for MSSA, none of the ATs successfully reduced MRSA biofilms, even at the highest concentration of AT compound. However, treatment success of biofilms may also be dependent on the strain of *Staphylococcus*, and we will continue to test AT anti-biofilm efficacy against other strains of *S. aureus* as well as biofilm-forming strains of *Staphylococcus epidermidis* [61].

Checkerboard assays with daptomycin and vancomycin were used to assess potential modes of action. For these assays, a computed FICi below 0.5 indicates synergy, whereas a score greater than 4 indicates antagonism. None of the ATs exhibited an antagonistic effect with daptomycin or vancomycin. All four ATs showed synergy with daptomycin against MRSA suggesting that these ATs target cell membranes [62]. Although many triazoles target fungal cell membrane synthesis, they can also exhibit other modes of action; thus, the daptomycin checkerboard assay is insufficient to draw conclusions about the anti-Staphylococcal mode of action for these molecules. Daptomycin acts by reorganizing Gram-positive bacterial membranes, leading to membrane permeabilization [62,63]. Membrane permeabilization could facilitate AT targeting of intracellular molecules. In fact, the bioavailability model of drug synergy proposes that two drugs can be synergistic if the action of one drug assists with another drug’s availability to target cells, such as by increasing the second drug’s entry into the cell [64]. We will investigate AT disruption of membrane integrity using phenotypic assays for bacterial cytoplasmic membrane damage, such as the LIVE/DEAD^®^ BacLight^TM^ Bacterial Viability Kit or similar assays [65,66,67]. Synergy with vancomycin was only evident with AT116 and AT125 against MSSA, indicating the cell wall as a possible target of these ATs. Other 1,2,3-triazole-containing antibiotics, including cefatrizine and tazobactam, also target bacterial cell walls [47,48,49,50]. It is interesting that AT106 had FICi values between 0.5–1 for MSSA and MRSA as it has been suggested that values of 0.5 < FICi < 1 indicate an additive to synergistic effect for antimicrobials [68]. Again, it is possible that the synergy could result from vancomycin allowing entry of AT compounds into the bacterial cells. Future studies will use aurone combinations and expand antibiotics used for checkerboard assays to assess whether intracellular processes such as nucleic acid synthesis, protein synthesis, or metabolic pathways are affected. We have already shown that aurone antifungal activity is related to both oxidative stress and interference with amino acid and carbohydrate biosynthesis, and these ATs may also have multiple modes of action [30].

The structures of these aurone-derived triazoles may also be important to their anti-Staphylococcal activity. In a study evaluating fluoro- and trifluoromethyl-substituted salicylanilide derivatives against *S. aureus* [69], the authors report the significance of the 2-hydroxyl group on the benzoic acid portion for anti-Staphylococcal activity. All of the ATs in our study have a similar salicyl functional array. The authors of the salicylanilide derivatives study also reported the significance of halogen substitutions, especially bromine and trifluoromethyl groups to target Staphylococci [69]. All four of the ATs in this study are halogenated, with AT106 (chlorine) and AT116 (trifluoromethyl) having the halogen on the triazole phenyl group. Indeed, in the case of AT116, the trifluoromethyl group is meta, exactly as it is found in the optimal salicylanilides examined by Lal et al. [69]. AT125 and AT137, which were both effective against MRSA, contain bromine, although they differ in the location of bromine as well as the type of aromatic ring that contains them. The AT137 bromine is on a thiophene ring attached to the triazole, whereas the bromine of AT125 is on the salicyl portion, similar to that observed in triazole-substituted chalcones reported by Kant et al. [49]. One of the halogenated compounds described by Lal et al. [69] also had significant antibiofilm activity, although, similar to the results seen for our ATs, that activity required a 10-fold increase in the concentration of the compound. Other halogen-containing polyphenols, especially brominated furanones, have also been identified as having anti-biofilm properties for Gram-positive and Gram-negative bacteria, including Staphylococci. The proposed mechanism of action for these furanones is as quorum-sensing inhibitors [70,71,72,73,74,75]. Although the AT compounds no longer contain the aurone furan ring, the AT ring structures are halogenated with both AT125 and AT137 having a bromine either on the salicyl or the thiophene ring. This suggests a potential mechanism of action for the ATs in disrupting quorum sensing that will be further investigated.

Our AT library has additional compounds with structures comparable to AT125 and AT137 that possess bromines in other positions or are substituted with another halogen (Appendix A). Although several of these were anti-Staphylococcal, they had lower SI indexes with mammalian cells and were not pursued; however, the structures of some of these compounds support the activity of the ATs in this study. ATs having halogens at the para position of the phenyl group had increased activity against *S. aureus* (AT106, AT113; Appendix A), with a para halogen being less active against MSSA, but similar against MRSA strains (AT107; Appendix A). Unlike AT137, AT135 (Appendix A) has no substitution on the triazole thiophene ring, and without bromine, the anti-MRSA activity decreased by over 15-fold. ATs with meta or para halogens on the salicyl ring had increased anti-Staphylococcal activity (AT119, AT120, AT122-AT126; Appendix A). These compounds all have a benzonitrile ring attached to the triazole, except for AT123 which has a methylbenzene ring (Appendix A). In all cases, the carbonitrile or methyl group is para. AT100 (Appendix A) also has a para methylbenzene ring on the triazole, but does not possess a halogen, yet exhibited significant inhibition against MRSA (>96%). Although the SI indexes were low, indicating toxicity for human cells, we plan to use these other ATs in phenotypic assays to determine modes of action for AT125 and AT137, as these compounds will help us to assess the structure–activity relationships between these molecules and their bioactivity. This expanded testing will provide a more comprehensive view of anti-Staphylococcal activity for the aurone-derived triazoles, which not only have the potential for multiple modes of antimicrobial activity but are also a cost-effective, explorable platform for antibiotic development.

## 4. Materials and Methods

### 4.1. Aurone-Derived Triazole Compounds

The syntheses and characterizations of the AT compounds used in this study have been previously described [34,76]. Purified powders of the AT compounds were solubilized in 100% dimethyl sulfoxide (DMSO) to a concentration of 10 mM. For bioactivity assays, these stock solutions were further diluted in Cation-Adjusted Mueller–Hinton Broth (CAMHB), RPMI-1640 (RPMI), or Brain Heart Infusion Broth (BHI) to achieve the final concentration of interest which ranged from 3.125 to 100 µM. AT stock solutions were stored at −20 °C.

### 4.2. Bacteria Culture Preparation

MRSA strain ATCC 43300 and MSSA strain *S. aureus* ATCC 29213 were grown on Tryptic Soy Agar (TSA) plates. Before use, an isolated colony was transferred from the TSA plate into Tryptic Soy Broth (TSB) and incubated 18–24 h in a humidified incubator at 37 °C. To adjust for a standard concentration of bacteria, the turbidity of the culture was measured at 600 nm, and its optical density (OD_600_) adjusted to be in the range 0.08–0.12 which equates to a concentration of approximately 1 × 10^8^ CFU/mL. This suspension was diluted 1:20 in TSB. The final mixture for experiments used the bacterial suspension and AT compound mixture as a 1:10 dilution which resulted in a final 5 × 10^5^ CFU/mL bacterial concentration [77]. MRSA strain *S. aureus* ATCC 43300 and MSSA strain *S. aureus* ATCC 29213 were used for all testing in this study.

### 4.3. Antibacterial Assay

The assay for anti-Staphylococcal activity of AT compounds was prepared in 96-well transparent, U-bottom plates. A series of 2-fold dilutions of AT concentrations was tested in triplicate, with each well containing 10 µL of prepared bacterium and 90 µL of AT mixture in CAMHB. This resulted in 100 µL of volume in each well with a 5 × 10^5^ CFU/mL bacterial concentration and 100, 50, 25, 12.5, 6.25, or 3.125 µM concentrations of the AT compound. Blank wells with only CAMHB, a negative control well with 10 µL of bacterium solution mixed into 90 µL CAMHB, a vehicle control well with bacteria mixed into 1% DMSO in CAMBH, and a positive cell death control with vancomycin (1 µL of stock 2 mg/mL solution in 89 µL CAMHB with 10 µL of bacteria) were included. The plate was incubated for 18–24 h at 37 °C in a humidified incubator. Before measurement, PrestoBlue^TM^ dye (Invitrogen, Carlsbad, CA, USA) was added (10 µL per well) and incubated for 1 h with aluminum foil covering the plate. A CLARIOstar^®^ (BMG LABTECH, Cary, NC, USA) multi-mode plate reader was used to measure the fluorescence at wavelengths of 560 nm for excitation and 590 nm for emission. Inhibitory concentration (IC_50_) values were defined as half inhibitory concentration and were computed using GraphPad Prism version 9.1 software [78].

### 4.4. Mammalian Cell Lines

Human HeLa epithelial (ATCC CCL-2) and HepG2 liver epithelial (ATCC HB-8065) cell lines were used to determine toxicity of AT compounds for mammalian cells. The cells were grown in a complete culture medium of RPMI 1640 with 10% heat-inactivated fetal bovine serum (FBS) at 37 °C in 5% CO_2_ (95% room air) in a humidified incubator. Before use, attached cell monolayers, which had reached at least 90% confluency, were detached by trypsinization with trypsin-EDTA (0.25%) and resuspension in fresh complete culture media. For experiments, the stock concentration of 1 × 10^5^ cells/mL (to achieve a final 1 × 10^4^ cells/well concentration) was achieved by counting the trypsinized cells and adjusting to their final concentration for assays.

### 4.5. Cytotoxicity Assay

HeLa or HepG2 cells were pre-seeded into a 96-well transparent, flat bottom plate by adding a volume of 100 µL stock cell concentration (1 × 10^5^ cells/mL) in complete culture media to each well. The plate was incubated overnight at 37 °C in a 5% CO_2_ (95% room air) humidified incubator. After the incubation and before mixing in the AT compounds, a 100 µL volume of fresh complete culture media was added to the pre-seeded wells. The 2-fold dilutions of AT compounds in triplicates were prepared on a separate 96-well plate. Once the dilution plate was prepared, dilutions were transferred into the pre-seeded plate to achieve the same concentrations of ATs as for the inhibition assay described above. The final inoculated plate was then returned to the 5% CO_2_ (95% room air) humidified incubator for an additional 24 h of incubation with the diluted ATs. Controls were blank wells containing only media, wells with cells and DMSO in complete culture media, and wells with cells and complete culture media only. Measurement of toxicity was analogous to the inhibition assay except that 20 µL of PrestoBlue^TM^ was used and the incubation was 3–6 h before reading with the plate-reader. Cytotoxicity concentration (CC_50_) was defined as the concentration where half of the cells remained viable and was computed using GraphPad Prism.

### 4.6. Hemolysis Assay

A single donor-specific blood sample was acquired from Innovative Research Inc. (Novi, MI, USA) for the assay (donor#:1WB3ALS40ML-36517). A volume of 1–2 mL of the human red blood cells (hRBCs in Alsevier’s solution) was mixed with 8–10 mL of Phosphate-Buffered Saline (PBS) in a 15 mL centrifuge tube to pellet by centrifugation at 1400× *g* in a swinging bucket centrifuge for 10 min. The supernatant was then removed, and the cells were washed with 10 mL of PBS two additional times. The cells were resuspended in PBS to achieve a final 1% solution of hRBCs.

The wells of a 96-well U-bottom plate that contained AT compounds were prefilled with 50 µL of PBS. A 2-fold dilution of AT compounds was prepared. For each triplicate, an initial AT dilution was prepared using 6 µL of AT stock solution in 144 µL of PBS. This initial mixture was diluted 2-fold on a 96-well plate following the method described above for a final volume of 50 µL of diluted AT in each well. As a final step, 50 µL of the 1% hRBC solution was added to each well, except the blank wells. PBS and hRBCs were used for negative control, whereas hRBCs with 1% and 0.1% Triton-X100 were used as positive controls. After 1 h of incubation at 37 °C in a humidified incubator, the plate was centrifuged at 1000 rpm for 10 min in a Thermo Sorvall Legend XTR centrifuge with a TX-750 swinging bucket rotor (Thermo Fisher Scientific, Waltham, MA, USA) to pellet intact cells. After centrifugation, 50 µL of supernatant was transferred from each well into a new 96-well U-bottom plate and hemolysis of the hRBCs detected in a plate-reader for absorbance at 405 nm. Hemolytic activity for all concentrations of the ATs was computed following this formula:(1)$$\%\;hemolysis=\frac{\left({OD}_{405nm}sample-{\;OD}_{405nm}negative\;control\right)}{\left({OD}_{405nm}positivecontrol-{OD}_{405nm}sample\right)}$$

### 4.7. Growth Curve Assay

For the growth curve assay, honeycomb plates (Growth Curves USA, Piscataway, NJ, USA) were used with TSB as the growth medium. Using bacterial cultures as described above, growth curve measurements were carried out in quintuplicates. For positive growth controls, wells with 10 µL of bacteria mixed into 90 µL of TSB were used. For quintuplicate treatment wells, 4-fold dilutions of AT compounds were used such that AT concentrations of 100, 25, 6.25, and 1.56 µM were tested. A 10 µL inoculum of bacteria prepared as described above was added to each dilution of the ATs. The growth in each well was measured in 15 min intervals for 48 h at 37 °C in the Bioscreen C instrument [79]. The results were processed and analyzed with Microsoft Excel (version 2201) and GraphPad Prism 9.3.1 software (GraphPad by Dotmatics, Boston, MA, USA).

### 4.8. Checkerboard Assay

This preparation for the checkerboard assay was analogous to the antibacterial assay described above in that the standard (5 × 10^5^ CFU/mL) concentration of bacteria in media was mixed with 2-fold dilutions of antibacterial agents to verify their inhibition potential [80,81]. For this assay, the highest concentration of an agent was either 100 µM for the ATs or the minimal inhibitory concentration (MIC_90_—the concentration of the agent with at least 90% inhibition of the bacteria) for the antibiotics. Control wells contained 10 µL of bacteria with 90 µL of CAMHB, and the negative control was bacteria in 1% DMSO and CAMHB. Incubation, staining with PrestoBlue^TM^, and assessing cell viability are as described for the antibacterial assay above.

### 4.9. Biofilm Assay

To prepare AT-treated cultures, overnight bacterial cultures were used (adjusted to 1 × 10^8^ CFU/mL) in BHI medium to grow biofilm in 35 mm MatTEK dishes with a 20 mm glass bottom well by adding 1500 µL of overnight bacteria per dish. The dishes were incubated at 37 °C in a humidified incubator for 48 h. The old media was refreshed by adding 1500 µL of BHI for the control, or the AT mixture (15 µL of AT stock with 1485 µL BHI to achieve 100 µM AT concentration), or DMSO mixture (15 µL DMSO with 1485 µL BHI) for negative control. The dishes were then incubated in a humidified incubator for an additional 24 h at 37 °C. The media was removed, and cells were washed twice with 500 µL PBS, followed by the addition of 50 µL BacLight^TM^ with 250 µL PBS for 15 min. Biofilms were imaged using a Zeiss AxioObserver microscope with an LSM700 confocal module (Carl Zeiss Microscopy, White Plains, NY, USA). To assess biofilm disruption, Fiji [82] software version 2.14.0 was used. Initially, raw microscope images were imported, separating the color channels, as dead cells were stained fluorescent red and living cells were stained green. To allow particle counts, the color saturation threshold was set using the RenyiEntropy option [83] to clear the images from potential noise. This step was followed by watershed separation [84], which separates potentially overlapping cells, and then particle analysis was completed using the 0.5–1.5 micron size as a guide. Biofilm disruption was quantified both by reductions in total bacterial cell numbers and increases in dead cell numbers.

## Figures and Tables

**Figure 1 antibiotics-12-01370-f001:**
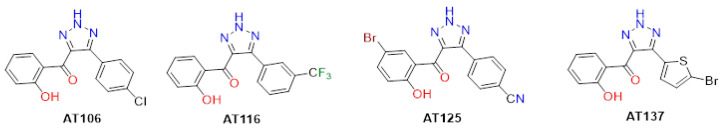
Chemical structure of aurone-derived triazoles AT106, AT116, AT125, and AT137.

**Figure 2 antibiotics-12-01370-f002:**
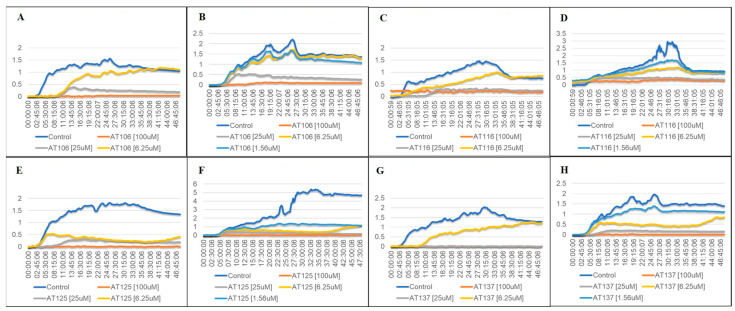
Growth curve measurements of MSSA or MRSA in the presence of 0–100 µM concentrations of the ATs. The 48 h incubation was ample time for bacteria to enter stationary or death phase. AT125 and AT137 exhibited the greatest inhibition against MRSA. (**A**) MSSA growth in AT106; (**B**) MRSA growth in AT106; (**C**) MSSA growth in AT116; (**D**) MRSA growth in AT116; (**E**) MSSA growth in AT125; (**F**) MRSA growth in AT125; (**G**) MSSA growth in AT137; (**H**) MRSA growth in AT137.

**Figure 3 antibiotics-12-01370-f003:**
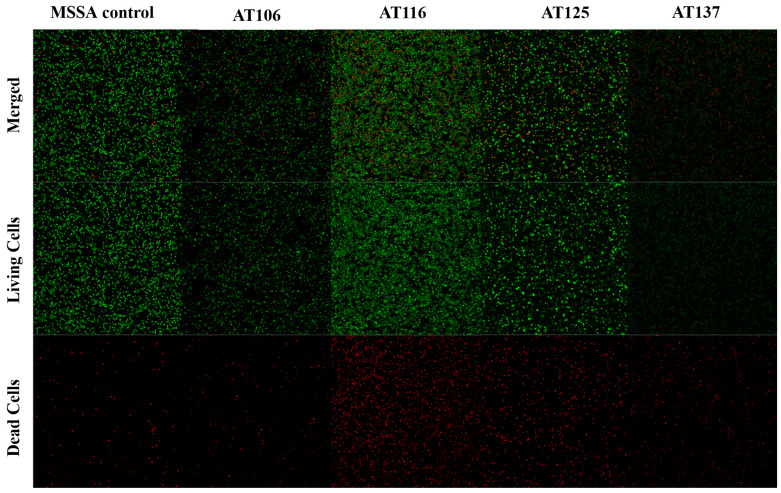
Confocal images of pre-formed, 48 h MSSA biofilms treated with AT compounds for 24 h. Following treatment, biofilms were stained with BacLight^TM^ (Invitrogen, Carlsbad, CA, USA), a SYTO9 and propidium iodide mixture, such that bacteria with intact cell membranes stain fluorescent green, whereas those with damaged membranes (dead or dying bacterial cells) stain fluorescent red. In the images above, the bottom row shows dead cells (red), the center row depicts living bacteria (green), whereas the top row is the merged image combining both live and dead cells.

**Table 1 antibiotics-12-01370-t001:** Half-maximal inhibitory concentrations (IC_50_) of AT compounds against MRSA and MSSA as well as half-maximal cytotoxicity levels (CC_50_) for HepG2 (ATCC HB-8065) and HeLa (ATCC CCL-2) cells. The computed column of SI represents the ratio of the MSSA IC_50_ to the HepG2 CC_50_.

AT	IC_50_ (µM)		CC_50_ (µM)		Max SI (CC_50_/IC_50_)
	MSSA	MRSA	HepG2	HeLa	
AT106	5.439	22.380	70.660	60.330	12.99
AT116	3.178	8.295	50.870	56.770	16.01
AT125	4.325	5.412	50.570	51.300	11.69
AT137	3.092	3.870	39.810	2.838	12.88

**Table 2 antibiotics-12-01370-t002:** Hemolytic activity of dilutions of AT compounds on human red blood cells (donor#:1WB3ALS40ML-36517). All ATs show little to no hemolytic activities at their IC_50_ concentrations.

Concentration (µM)	Hemolysis by AT Compounds (%)			
	AT106	AT116	AT125	AT137
100	1.1	3.5	5.1	1.8
50	−0.3	2.6	4.9	1.4
25	1.2	2.1	3.9	2.7
12.5	1.2	2.3	2.3	−0.8
6.25	−0.1	−0.7	3.6	−0.5
3.125	0.6	0.8	0.9	−2.6

**Table 3 antibiotics-12-01370-t003:** FICi of ATs (concentrations range from 64 µM to 1 µM) with vancomycin and daptomycin in checkerboard assays. The smallest ratio is represented. FICi = IC_a_/MIC_a_ + IC_b_/MIC_b_, where IC is the inhibitory concentration in the mix and MIC is the minimum inhibitory concentration of that compound alone. Values below 0.5 suggest synergy, whereas values above 4 indicate antagonism between compounds. Values between 0.5–4 indicate indifference or no interaction.

FICi	Vancomycin		Daptomycin	
	MSSA	MRSA	MSSA	MRSA
AT106	0.57	0.52	0.25	0.05
AT116	0.25	1.03	0.28	0.05
AT125	0.16	1.02	1.18	0.38
AT137	1.5	1.13	0.38	0.19

## Data Availability

Not applicable.

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
