# Peer review of "Evaluation of the Antibacterial Effect of Aurone-Derived Triazoles on Staphylococcus aureus"

_antibiotics, 2023, doi:10.3390/antibiotics12091370_

Round 1

Reviewer 1 Report

The authors studied the antibacterial activity and cell toxicity of some new pharmaceutical compounds from the aurone-triazoles class. They used growth-curve analysis, biofilm inhibition microscopy, and checkerboard assay to assess the antibacterial activity and toxicity for mamallian cells.

The study is generally well written and thorough. 

Lines 120-125. Please explain for readers unfamiliar with the biofilm assays which compounds had effects against bacterial biofilms and how can you deduce this from the image. In the figure caption is stated that the effect against biofilm is based on the number of dead cells, this should be stated also in the text. How do you interpret the much lower number of dead and living cells in compound AT137? Does this mean it disrupted the bacterial biofilm?

Reviewer 2 Report

Dear authors, I find the article interesting and within the theme of the journal; however, I believe that some modifications must be made to be considered for publication: 

Abstract. 

Add the results referring to biofilms and hemolytic activity. 

Introduction. 

Information about biofilms can be added, what they are, and their impact in the clinical area. 

Results 

Line 71-75 move to the discussion section 

Line 135-136-move to the discussion section

Section 2.3. Write the results more clearly, since they are confusing. Section 2.4. Write the results more clearly, since they are confusing. Line 138-139. Many of the values are greater than 0.5; this indicates that there is no synergism according to the scale?. (below 0.5 suggest synergy). The analyzed compounds present structural similarities to confirm that they have the same antimicrobial mechanism of action?.

 Discussion 

The discussion should be improved since the results are not discussed in detail, and comparative analyses are not carried out clearly. 

There is no discussion about cytotoxic activity and hemolysis. 

There is no discussion about  biofilms. How do the compounds affect the performed biofilms?. Try to explain the mechanism on biofilms. 

The live/dead assay discussion needs to be clarified since the assay was performed on biofilms. It would help if you tried to differentiate the effect against planktonic cells and biofilms since they are different.

The obtained result in MIC evaluation is good? Use an established scale if it is possible. 

Compare with reference antibiotics to know how active they are.

Author Response

Please see attached attachment.

Round 2

Reviewer 2 Report

Dear authors

 Most of the observations were addressed correctly. I suggest improving the biofilm topic in the introduction and improving the discussion on biofilms (suggest the mechanism of action of tested compounds). 

Regards.  
